# Pattern Matching-based Out-of-Distribution Detection for Multi-label Node Classification

## Abstract

Graph neural networks (GNNs) have achieved dominant performance in various prediction tasks on graphs. When deploying GNNs in the real world, estimating the possibility of out-of-distribution (OOD) testing samples becomes a crucial safety concern. Although some research has investigated the graph OOD detection problem, most have concentrated on single-label classification scenarios, a specific case of the more general multi-label classification, which has broader applications, such as in social networks where nodes can represent users with multiple interests or attributes. In this paper, we first introduce and define the multi-label graph OOD detection problem and propose a simple yet effective pattern matching-based OOD detection method to address it. In particular, our method utilizes feature pattern matching and label pattern matching to obtain two matching scores. By incorporating topological structure adjustment, we ultimately derive confidence scores, serving as indicators of the likelihood that a test sample is an OOD instances. We conduct extensive comparisons with existing OOD detection methods in the context of multi-label graphs. The results show that our method achieves an impressive 7.61% reduction in FPR95 compared to the leading baselines, setting a new state-of-the-art. Furthermore, our approach can serve as a benchmark for OOD detection on multi-label graphs.

## 1 Introduction

Graphs serve as a fundamental data structure that plays a pivotal role in modeling intricate relationships between entities, including social, biological, and internet networks. With the rapid advancement of deep learning, the emergence of graph neural networks (GNNs) (Scarselli et al., 2008) has revolutionized the way of analyzing and leveraging graph-structured data (Taghibakhshi et al., 2023; Brilliantova et al., 2023; Gao et al., 2021). GNNs are proven to be exceptionally adept at uncovering complex patterns and making predictions based on the graphical inter-connections (between nodes and edges) along with their associated features, enabling graph learning tasks, like node classification, link prediction, and more.

However, when deploying a trained GNN involving graph-structured data in practice, it is inevitable to encounter previously unseen out-of-distribution (OOD) test inputs, which do not conform to the distribution of the training data. While graph OOD detection is underscored by recent research (Zhao et al., 2020; Stadler et al., 2021; Wu et al., 2023), it is worth highlighting that existing studies primarily focus on *single-label classification, where each node is assigned to a single label*. Notably, the more general and practical scenario of *multi-label graphs* remains largely unexplored.

Basically, in single-label classification tasks on graphs, each node is assigned an exclusive label from a predefined set of classes. For example, as shown in Figure 1 (a), in the analysis of occupational categories of users in social networks, nodes might be categorized as teachers, engineers, or doctors, with each node belonging to only one of these categories. In contrast, in a multi-label graph, nodes can simultaneously belong to multiple categories. As shown in Figure 1 (b), when analyzing users' interests and hobbies, each user can have multiple attributes, such as music, sports, and movies. In this case, a user can be associated with multiple labels simultaneously. Compared to the former,

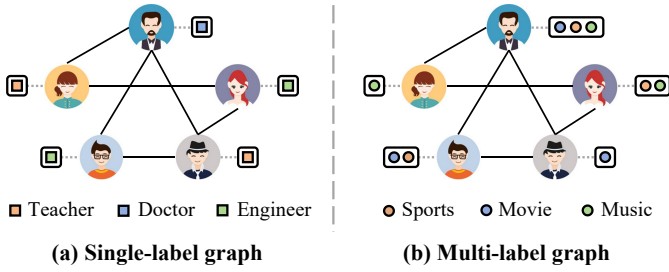

Figure 1: A social network graph is depicted, where nodes represent users, and edges represent relationships between users. In (a), the labels correspond to users' occupations, while in (b), the labels represent users' interests and hobbies.

the latter is more general and flexible, finding applications in a broader range of scenarios, and presenting greater technical challenges.

A proliferation of research on OOD detection methods has emerged for both single-label and multi-label problems. These single-label methods (Hendrycks & Gimpel, 2017; Liang et al., 2018; Wang et al., 2022; Zhao et al., 2020; Stadler et al., 2021; Wu et al., 2023) primarily focus on graph OOD detection in scenarios where each node is assigned a single label. Thus, they typically only need to consider information from individual labels and do not need to account for dependencies between them. However, existing research (Wang et al., 2021) has demonstrated that for multi-label OOD detection problems, integrating information from multiple labels is crucial. Additionally, these detection methods can be further categorized based on the inputs type, including graph or non-graph data (e.g., images), while the non-graph inputs are typically assumed to be *i.i.d* sampled, the relational structure in graph induces data inter-dependence, necessitates the consideration of graph structure for OOD detection while hindering trivial adaptation of non-graph based methods(Wu et al., 2023). On the other hand, the focus of researches in multi-label OOD detection resides on non-graph data (Wang et al., 2021; Hendrycks et al., 2022). Apart from the aforementioned drawback of neglecting graph structure, these methods relies solely on the logits, which limits their performance due to the singleness of the information source (Wang et al., 2022). In contrast, integrating information from multiple sources, such as feature or softmax probability, has proven effective for OOD detection (Wang et al., 2022). Table 1 summarizes the detection categories discussed above, along with the representative methods. In a nutshell, existing OOD detection methods are inapplicable to graph-based multi-label scenarios, which motivates us to fill this gap.

In this paper, we formulate the problem of OOD detection for multi-label graph node classification. To contend with the aforementioned technical challenges, we propose a simple yet effective pattern matching-based OOD detection method tailored for multi-label graphs. Specifically, our method initiates by transforming the graph data into feature and logit spaces

Table 1: Category and comparison of OOD Detection Methods. The inability of existing methods to capture critical and comprehensive information prevents them from being applied to graph-based multi-label scenarios.

| | | Method | Information | | |
| --- | --- | --- | --- | --- | --- |
| | | | multiple label | grpah structure | multiple sources |
| **Single label** | **non-graph** | VIM | × | × | ✓ |
| | **graph** | GNNSAFE | × | ✓ | × |
| **Multi label** | **non-graph** | JointEnergy | ✓ | × | × |
| | **graph** | OURS | ✓ | ✓ | ✓ |

via a classical GNN model. Then, we define feature patterns based on the distribution of training samples, highlighting the differences between ID and OOD samples in feature distributions. Additionally, to utilize label correlation dependencies inherent in multi-label nodes, we define label patterns from a label dependency matrix. During the test phase, we match the feature ang logits of samples with these patterns, calculating the feature matching score and label matching score to address the differences in feature and label distributions between training and the test samples. Afterwards, we calibrate these two scores, using the graph's topological structure via a classical label propagation algorithm to derive a confidence score, which determines whether a sample is OOD. Despite its simplicity, our method effectively differentiates OOD samples from test inputs, by integrating pattern information from the training data with the graphical structure characteristics.

Our contributions can be summarized as follows.

- We formulate the problem of multi-label graph OOD detection, which, while important, has been previously unexplored.

- To tackle the challenges specific to multi-label graph OOD detection, we propose a solution based on pattern matching, which aims at utilizing feature distributions and label dependencies of the training samples to obtain the pattern and label matching scores. Additionally, we adjust these two scores using the topological structure of the graph to increase the score disparity between ID and OOD samples.

- We perform comprehensive experiments on OOD detection using multi-label graph datasets spanning diverse domains. The experiment results demonstrate the significant superiority of our approach over existing detection methods. For instance, on the Yeast dataset, our method achieves an impressive 7.15% reduction in the false positive rate (at 95% FPR) compared to the state-of-the-art baselines. Furthermore, consistent performance enhancements were observed across various multi-label tasks and diverse network architectures.

## 2 RELATED WORK

### 2.1 MULTI-LABEL NODE CLASSIFICATION

Multi-label node classification refers to the task of assigning multiple labels to each node in a graph. This has widespread applications in the real world, such as protein function prediction (Zhao et al., 2023), social networks (Wu et al., 2018), and more (Xu et al., 2013). The methods for addressing the task of multi-label node classification can be categorized into three main types.

The first category involves methods based on node embedding. For example, (Perozzi et al., 2014; Khosla et al., 2020) employs a lookup table to generate embeddings where similar nodes are closely located. The learned representations are then applied as input features to various downstream prediction modules.

The second category involves methods based on convolutional neural networks, such as (Shi et al., 2020; Zhou et al., 2021). These methods commence by gathering node representations through the aggregation of feature information within their local neighborhoods. Then, the extracted feature vectors are fused with label embeddings to form ultimate node embeddings. Subsequently, these node embeddings are fed into a classification model for the generation of node labels.

The last category is based on graph neural networks, with the differences primarily in how the aggregation layers are implemented. The classical Graph Convolutional Network (GCN) (Kipf & Welling, 2017) performs degree-weighted aggregation over neighborhood features. Since GCN is limited in learning general neighborhood mixing relationships, MixHop (Abu-El-Haija et al., 2019) proposes learning these relationships by repetitively mixing features of neighbors at varying distances. Additionally, some methods combine GNNs with label propagation (LPA) algorithm. For instance, GCN-LPA (Wang & Leskovec, 2020) utilizes LPA as regularization to assist GCN in learning appropriate edge weights, thereby enhancing classification performance.

Instead, in this paper, we focus on the OOD detection problem in the context of multi-label graph node classification, which is underexplored.

### 2.2 OUT-OF-DISTRIBUTION DETECTION

In recent years, the problem of OOD detection has garnered widespread attention, particularly in the context of vision tasks. A category of methods (Hendrycks & Gimpel, 2017; Liang et al., 2018) relies on computing uncertainty scores based on the output of trained neural network, to measure OOD samples. For example, (Hendrycks & Gimpel, 2017) proposes that samples with lower maximum softmax probability values in model predictions are more likely to be OOD. Also, there are OOD detection methods based on distances between samples in the feature space (Sehwag et al., 2021; Sun et al., 2022), as well as methods involving training a generative model for OOD detection (Ren et al., 2019). Given that previous methods primarily focused on detecting OOD samples in the setting of multi-class classification, where each sample is assigned to a single label, some methods(Wang et al., 2021; Hendrycks et al., 2022) have been proposed to address the challenge of multi-label OOD

detection. For example, (Wang et al., 2021) estimates an OOD uncertainty score by aggregating label energy scores from multiple labels.

However, as the aforementioned methods are primarily designed for vision tasks, most of them rely on the assumption that input samples are independently and identically distributed, which is evidently not applicable to the scenario of node classification in graph (Wu et al., 2023). Consequently, some OOD detection methods (Zhao et al., 2020; Stadler et al., 2021; Wu et al., 2023) tailored for graphs have been proposed. For example, (Zhao et al., 2020) proposes a Graph-based Kernel Dirichlet Distribution Estimation(GKDE) method, aiming to accurately predict node-level Dirichlet distributions and detect OOD nodes. (Wu et al., 2023) utilizes an energy function directly extracted from a GCN to discriminate whether an input sample belong to ID or OOD.

The existing graph-based OOD detection methods mainly focus on single-label node classification task where each node is assigned with only one label, overlooking the crucial application scenario of multi-label node classification in graph.

## 3 PROBLEM SETUP

### 3.1 MULTI-LABEL GRAPHS

Consider an undirected graph $G = \{V, E\}$, where $V = \{v_1, \ldots, v_N\}$ represents the set of vertices (or nodes), with $N$ denoting the number of vertices and $E = \{e_1, \ldots, e_M\}$ represents the set of edges connecting these vertices. Let $\mathbf{A} \in \{a_{i,j}\}_{i \leq N, j \leq N}$ be the adjacency matrix of the graph, where each element $a_{ij} = 1$ indicates the existence of an edge connecting $v_i$ and $v_j$, and 0 otherwise. In a multi-label setting, each vertex $v_i$ is associated with a label vector denoted as $\mathbf{y}_i = \{y_1, y_2, \ldots, y_{K_l}\}$, where $K_l$ is the total number of classes.

### 3.2 MULTI-LABEL NODE CLASSIFICATION (MLNC)

Given a graph $G$, following the semi-supervised learning paradigm, we assume that there are $N_l$ labeled nodes forming the training set $D_{tr}$, and $N_u$ unlabeled nodes forming the testing set $D_{te}$, where $N = N_l + N_u$. The goal of multi-label graph node classification is to learn a mapping $f : (V, A) \to \hat{\mathbf{Y}}$, where $\hat{\mathbf{Y}} = \{\hat{\mathbf{y}}_1, \ldots, \hat{\mathbf{y}}_N\}$ is the predicted label matrix, and $f(\cdot)$ is a graph neural network.

### 3.3 OOD DETECTION FOR MLNC

The task of multi-label graph node classification aims to achieve high classification performance on test nodes, under the assumption that test inputs have identical distribution with training data, referred to as the in-distribution (ID). A robust classifier for multi-label graph node classification should not only accurately classify ID inputs (from known categories) but also effectively identify OOD test inputs (from unknow categories).

To formalize the problem, we consider two distributions $\mathcal{P}_{in}$ and $\mathcal{P}_{out}$ as ID and OOD data spaces, respectively. We operate under the assumption that training data $D_{tr}$ originate from $\mathcal{P}_{in}$. Given a set of unlabeled test inputs $D_{te}$, we assume it consists of both ID and OOD nodes, sampled from $\mathcal{P}_{in} \times \mathcal{P}_{out}$. Then, the goal of MLNC OOD detection is essentially to find a decision boundary $\mathcal{H}$ for any given test input $\mathbf{x}$, such that,

$$\mathcal{H}(\mathbf{x}, G_{\mathbf{x}}, \mathbf{A}) = \begin{cases} 1, & \text{if } \mathbf{x} \in \mathcal{P}_{in} \\ 0, & \text{if } \mathbf{x} \in \mathcal{P}_{out} \end{cases}$$

where $G_{\mathbf{x}}$ denotes the ego-graph centered at node $\mathbf{x}$. We adhere to the practice of utilizing the graphical topology to facilitate OOD detection during the testing phase, which contrasts with the OOD detection setups commonly used in vision tasks (Hendrycks & Gimpel, 2017), where test inputs are generally treated as independent instances.

# 4 METHOD

In this section, we introduce our proposed pattern matching strategy and label propagation algorithm in sections 4.1 and 4.2, respectively. The overall framework is shown in Figure 2.

## 4.1 PATTERN MATCHING

For multi-label classification tasks, it is conventional to append a sigmoid layer to a feature mapping backbone. The sigmoid function transforms the model's output logits into class probabilities.

$$p(y_k|v_i, G_{v_i}, \theta) = \frac{e^{f_k(v_i, G_{v_i}, \theta)}}{1 + e^{f_k(v_i, G_{v_i}, \theta)}} \tag{1}$$

Here, $p(y_k|v_i, G_{v_i}, \theta)$ denotes the predicted probability of node $v_i$ belonging to class $k \in [1, K_l]$, and $f_k(\cdot)$ represents the $k$-th element in the logit vector. Although the sigmoid function can intuitively indicate the probability of a node belonging to a specific category in multi-label classification tasks, directly applying the mapped probability values makes OOD detection more challenging. The sigmoid function compresses the real-number logit space into a $[0, 1]$ range, reducing the discrimination between ID and OOD nodes and making their separation difficult. As shown in Figures 3 (a) and (b), utilizing the maximal logit yields greater separation between ID and OOD nodes compared to using the maximal predicted probability. To generate the feature and logit vector, we employ a widely adopted Graph Convolutional Network (GCN) as the backbone, which iteratively aggregates neighbor features to update central node embeddings. GCN captures node inter-dependencies through convolution layers and layer-wise normalized feature propagation:

$$Z^{(l)} = \sigma(D^{-1/2}\tilde{A}D^{-1/2}Z^{(l-1)}W^l)$$
$$Z^{(l-1)} = [z_i^{(l-1)}]_{i \in V}, \quad Z^{(0)} = \mathbf{X}, \tag{2}$$

where $\tilde{A}$ is an adjacency matrix with a self-loop based on $A$, $D$ is its associated diagonal degree matrix, $W^{(l)}$ is the weight matrix at the $l$-th layer, and $\sigma$ represents a non-linear activation function. With $L$ layers of graph convolution, the GCN model outputs a $K_f$-dimensional feature vector $z^{(L-1)}$ and $K_l$-dimensional logit vector $z^{(L)}$ for each node, indicated as $f(v_i, G_{v_i}, \theta) = z_i^{(L)}$, where $\theta$ denotes the training parameter set of the GCN model.

### 4.1.1 FEATURE PATTERN MATCHING

Although feature space information has been widely leveraged in image-based OOD detection, it remains largely underutilized in graph-based OOD detection. Due to the higher dimensionality, representations in the feature space inherently preserve more information than those in the logit space. We define the feature distribution of the training data as the feature pattern and introduce a feature pattern matching score $s_f$ to quantify how test data aligns with this distribution. Motivated by NAC(Liu et al., 2023a), we derive $\mathbf{s_f}$ from the probability density function (PDF) of training data, denoted as $\delta_{in}$.

To avoid the influence of excessively large feature values that could overshadow the contributions of other relevant features, we introduce a lower bound parameter $r$ to truncate the PDF. Specifically, given the $k$-th feature value $z_k$ of a test sample and the $k$-th PDF of the training dataset $\delta_{in}^k(\cdot)$, the function for $\mathbf{s_f}$ is defined as follows:

$$\Phi_{in}^k(z_k; r) = min(\delta_{in}^k(z_k), r) \tag{3}$$

For simplicity, we employ a histogram-based method to model the probability density functions, dividing the feature values into $P$ intervals $\{I_1, I_2, \ldots, I_P\}$ on a logarithmic scales. Accordingly, the function for $\mathbf{s_f}$ can be rewritten as:

$$\Phi_{in}^k(z_k; O) = \frac{1}{N_{tr}} \min(count(I_j|z_k \in I_j), O) \tag{4}$$

Here, $N_{tr}$ denotes the total number of samples in the training set, and $count(I_j)$ represents the number of training samples whose $k$-th feature value falls within the interval $I_j$. The condition

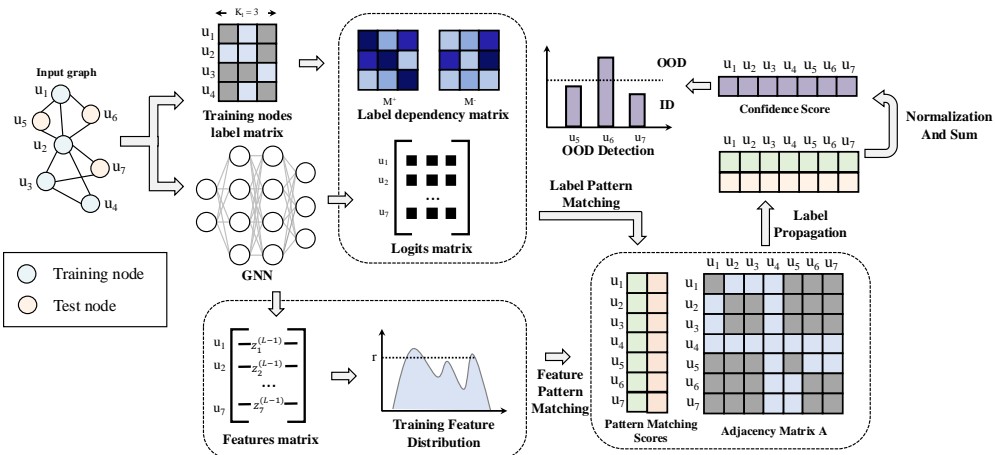

Figure 2: **Overview of the framework.** The GNN backbone processes the input graph data to produce a feature matrix $\mathbf{Z}^{(L-1)}$ and a logit matrix $\mathbf{Z}^{(L)}$. Two types of label dependency matrices, $\mathbf{M}^+$ and $\mathbf{M}^-$, are derived from the label matrix of the training data to capture label patterns, while a probability density function formalizes the feature distribution of the training data as the feature pattern. Next, the logit and feature information are used to compute label pattern matching scores and feature pattern matching scores, respectively. These scores are then adjusted using the adjacency matrix to enhance the distinction between ID and OOD samples. Finally, the scores are normalized and combined to produce a final confidence score, which determines whether a sample is classified as ID or OOD.

$z_k \in I_j$ ensures that the specific interval $I_j$ is selected based on the feature value $z_k$. This way, the function of feature pattern matching score $\mathbf{s_f}$ can be given as:

$$\mathbf{s_f} = \frac{1}{K_f} \sum_{k=1}^{K_f} \Phi_{in}^k(z_k; O) \tag{5}$$

The underlying intuition is that if the feature values of a test sample fall within the low-frequency regions of the feature probability distribution from the training data, it is more likely to be an OOD sample. Conversely, if the feature values fall within the high-frequency regions, the sample is more likely to be ID.

### 4.1.2 LABEL PATTERN MATCHING

In this section, we introduce the another pattern matching strategy, called the label pattern matching. In many existing multi-label learning works (Zhou et al., 2021; Chen et al., 2019), label correlation dependencies are frequently utilized to improve multi-label prediction tasks. These methods empirically support the notion that statistics derived from the labels in the training dataset can be leveraged to assess label correlation dependencies. Inspired by this, we propose a data-driven method that models the label dependencies in the training data as a label pattern. By matching the label structure associated with the test node's logits to this label pattern, we compute a label pattern matching score, which helps determine whether the node belongs to ID or OOD.

Specifically, we define positive dependency as the probability of the occurrence of the $i$-th label given the occurrence of the $j$-th label, denoted as $p(y_i = 1|y_j = 1)$. This captures the co-occurrence statistic between label pairs. Conversely, negative dependency is defined as the probability of the occurrence of the $i$-th label given the absence of the $j$-th label, denoted as $p(y_i = 1|y_j = 0)$. This captures the degree of independence between label pairs. By computing the frequencies of both patterns, we construct two corresponding label dependency matrices, denoted as $M^+$ and $M^-$:

$$\mathbf{M}_{ij}^+ = \frac{C_{(y_i=1\&y_j=1)}}{C_{y_j=1}} \quad \text{and} \quad \mathbf{M}_{ij}^- = \frac{C_{(y_i=1\&y_j=0)}}{C_{y_j=0}}, \tag{6}$$

where $C$ represents the counted frequency. Notice that the matrices are asymmetrical, as $p(y_i|y_j) \neq p(y_j|y_i)$. After obtaining the two matrices $M^+$ and $M^-$, to represent the label pattern, we propose

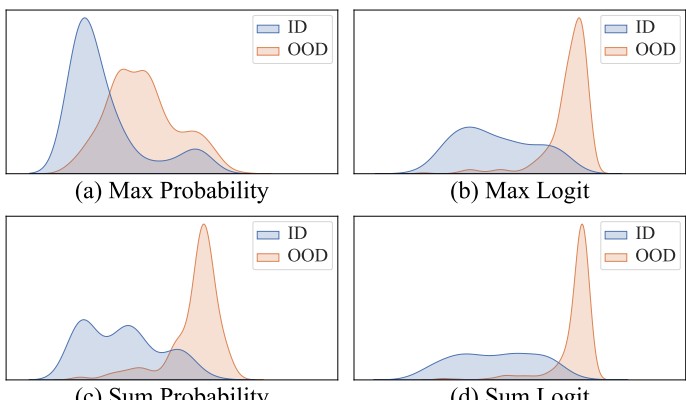

(a) Max Probability             (b) Max Logit

(c) Sum Probability            (d) Sum Logit

Figure 3: OOD detection using two kinds of score function (Probability vs. Logit) for multi-label node classification. Summing the logits leads to a greater difference in scores and improved separation between ID and OOD inputs, thereby enhancing OOD detection performance.

to build decision boundary in the logit space using a threshold $\tau$. Actually, the threshold $\tau$ can be determined based on the classification threshold used for the sigmoid function in MLNC, due to the function's monotonicity property.

$$\mathbf{e}_k = \begin{cases} \mathbf{z}_k^{(L)}, & \text{if } \mathbf{z}_k^{(L)} >= \tau \\ 0, & \text{if } \mathbf{z}_k^{(L)} < \tau \end{cases} \tag{7}$$

For each truncated logit vector $\mathbf{s}$, we extract rows from the corresponding matrix based on indices of the elements in the vector. This results in a new label dependency matrix, denoted as $\mathbf{M}$, where

$$\mathbf{M}_{ij} = \begin{cases} \mathbf{M}_{ij}^+, & \text{if } \mathbf{e}_j > 0 \\ \mathbf{M}_{ij}^-, & \text{if } \mathbf{e}_j = 0 \end{cases} \tag{8}$$

Subsequently, we get the label pattern matching score $\mathbf{s_l}$ based on the label pattern matrix $\mathbf{M}$ and the truncated logit vector $\mathbf{e}$.

$$\mathbf{s_l} = \sum_j (\mathbf{e} \cdot \mathbf{M}) \tag{9}$$

It is noteworthy that, compared to the approach using the maximum logit for OOD detection as proposed in prior works (Hendrycks et al., 2022), summing the logits yields better discrimination between ID and OOD nodes, resulting in a more distinguishable score (see Figures 3 (b) & (d)).

## 4.2 LABEL PROPAGATION

In this section, we leverage graphical topology to refine the feature pattern matching scores $\mathbf{S_f}$ and label pattern matching scores $\mathbf{S_l}$, via the classical label propagation algorithm. This technique is commonly employed for error correction and subsequent refinement of final predictions (Huang et al., 2021). Label propagation operates on the premise of local similarity, suggesting that neighboring nodes are more likely to share the same labels. This assumption is consistent with the observation that ID nodes tend to connect with other ID nodes, while OOD nodes are prone to connect with other OOD nodes (Liu et al., 2023b). By employing the label propagation algorithm, the pattern matching score of each node can be propagated to other nodes connected along connected path, resulting in calibrated pattern matching scores across neighboring node. This process improves the accuracy of detection of OOD nodes. To implement this, we employ an iterative propagation approach to refine the scores $\mathbf{S_f}$ and $\mathbf{S_l}$ as follows.

$$S_l^{(T)} = \alpha S_l^{(T-1)} + (1 - \alpha)D^{-1}AS_l^{(T-1)}$$
$$S_f^{(T)} = \alpha S_f^{(T-1)} + (1 - \alpha)D^{-1}AS_f^{(T-1)} \tag{10}$$

$$S^{(T)} = \alpha S^{(T-1)} + (1 - \alpha)D^{-1}AS^{(T-1)} \tag{11}$$

Here, $\alpha \in [0, 1]$ determines the balance between the node's own matching score and the scores of its connected neighbors, with the initial pattern matching scores represented as $S_f^{(0)}$ and $S_l^{(0)}$. With $T$ iterations of propagation, the refined pattern matching scores are obtained. After normalizing both scores to ensure they are within the same range, we combine them to generate the ultimate confidence score.

$$\tilde{S} = S_l + S_f \tag{12}$$

Finally, the confidence score is used to determine whether the test input is classified as ID or OOD.

## 5 EXPERIMENT

### 5.1 SETUP

**Datasets and Splits** The datasets used to validate our proposal are required to be graph-structured and multi-labeled. Following the setting in literature (Zhou et al., 2021), our experiments are based on five well-established real-world datasets[1] that serve as standard benchmarks for node classification: Facebook, Yeast, Movie, DBLP and BlogCatalog.

Table 2: Statistics of 5 benchmark datasets: node count, edge count, node attribute dimension, and distinct node label count.

| Dataset | Nodes | Edges | Attributes | Labels | ID Labels | OOD Labels | ID Nodes | OOD Nodes |
|---------|-------|-------|-----------|--------|-----------|-----------|----------|-----------|
| Facebook | 792 | 14,024 | 319 | 17 | 9 | 8 | 119 | 318 |
| Yeast | 1,240 | 1,674 | 370 | 13 | 4 | 9 | 149 | 647 |
| Movie | 7,155 | 404,241 | 5,297 | 20 | 13 | 7 | 1,704 | 340 |
| DBLP | 28,072 | 68,335 | 300 | 4 | 3 | 1 | 5,631 | 6,180 |
| BlogCatalog | 10,312 | 333,983 | 128 | 39 | 25 | 14 | 2,319 | 1,037 |

In line with prior research (Wang et al., 2021), we view a node as OOD node if it lacks any ID labels. Conversely, a node containing any ID labels is deemed as ID. For the ID nodes, 75% are randomly selected to serve as training nodes, while the remaining nodes are designated as ID test nodes. The statistics of datasets are presented in Table 2.

**Baselines** We compare our approach to three categories of OOD detection methods. The first category includes baselines focused on multi-class OOD detection in vision assuming i.i.d. input samples, and comprises MSP (Hendrycks & Gimpel, 2017) and ODIN (Liang et al., 2018). The second category includes baselines designed for multi-label OOD detection in vision, including JointEnergy (Wang et al., 2021) and MaxLogit (Hendrycks et al., 2022). The third category consists of baselines explicitly tailored for handling OOD data in graph learning, with GNNSAFE (Wu et al., 2023) serving as the SOTA method.

**Implementation Details** We set the iterations of propagation $T$ to 1 and the weight parameter $\alpha$ to 0.5. For a fair comparison, we employ the MixHop (Abu-El-Haija et al., 2019) as the backbone with a layer depth of 3, hop number of 2, and hidden size of 256. Additionally, we include other models, namely MLP, GCN (Kipf & Welling, 2017), GAT (Velickovic et al., 2017), and JKNet (Xu et al., 2018), for further discussions in our ablation study. More, we set the learning rate to 1e-4, the weight decay factor to 5e-4, and the dropout probability to 0.5. We reproduce all baselines following the settings of original papers with adaptation to fit our experimental setup.

**Evaluation Metrics** We use the following metrics: (1) the area under the receiver operating characteristic cureve (AUROC), (2) the area under the precision-recall curve (AUPR), and the false positive rate (FPR) of OOD samples, when the true positive rate of ID samples is at 95%.

### 5.2 MAIN RESULTS

**Performance Comparison.** We present the experimental outcomes of our approach in comparison with competitive methods in Table 3. The results consistently demonstrate that our method outperforms all competitors across all datasets. Notably, our method exhibits a significant 7.61% reduction

---

[1]The details of datasets are described in supplementary material.

Table 3: **Performance comparison with baselines.** The best performing method is in bold, the second best is underlined.

| Method | Facebook FPR ↓ | Facebook AUROC ↑ | Yeast FPR ↓ | Yeast AUROC ↑ | Movie FPR ↓ | Movie AUROC ↑ | DBLP FPR ↓ | DBLP AUROC ↑ | BlogCatalog FPR ↓ | BlogCatalog AUROC ↑ | Average FPR ↓ | Average AUROC ↑ |
|---|---|---|---|---|---|---|---|---|---|---|---|---|
| MSP | 18.87 | 94.74 | 81.45 | 76.08 | 76.76 | 61.32 | 80.05 | 81.13 | 91.42 | 69.71 | 76.18 | 77.40 |
| ODIN | 38.99 | 94.47 | 86.71 | 74.29 | 92.69 | 56.65 | 90.49 | 70.44 | 91.61 | 67.23 | 80.10 | 72.62 |
| Max Logit | 99.37 | 84.72 | 92.27 | 75.30 | 96.47 | 55.82 | 58.54 | 87.17 | 91.90 | 66.20 | 87.71 | 73.84 |
| JointEnergy | 98.43 | 32.77 | 94.28 | 74.05 | 95.59 | 43.61 | 56.52 | 88.14 | 92.77 | 62.94 | 87.52 | 60.30 |
| GNNSAFE | 11.95 | 93.74 | 87.64 | 70.88 | 67.11 | 70.09 | 45.24 | 87.90 | 90.94 | 67.67 | 61.33 | 78.54 |
| Ours | **7.55** | **95.15** | **74.50** | **78.79** | **56.76** | **71.17** | **43.41** | **89.34** | **86.40** | **69.83** | **53.72** | **80.86** |

Table 4: Performance comparison on varying OOD labels in DBLP. Each label, such as database (DB), is individually designated as the OOD label, while the remaining three labels serve as the ID labels.

| Method | DB FPR ↓ | DB AUPR ↑ | DB AUROC ↑ | DM FPR ↓ | DM AUPR ↑ | DM AUROC ↑ | AI FPR ↓ | AI AUPR ↑ | AI AUROC ↑ | IR FPR ↓ | IR AUPR ↑ | IR AUROC ↑ |
|---|---|---|---|---|---|---|---|---|---|---|---|---|
| MSP | 80.05 | 77.74 | 81.13 | 77.09 | 62.00 | 77.21 | 72.50 | 89.47 | 82.96 | 72.75 | 72.85 | 80.46 |
| ODIN | 90.49 | 65.73 | 70.44 | 89.90 | 48.82 | 68.33 | 88.83 | 81.03 | 72.10 | 88.70 | 59.90 | 71.77 |
| Max Logit | 58.54 | 86.15 | 87.17 | 63.36 | 71.45 | 82.20 | 52.80 | 92.82 | 87.62 | 58.36 | 80.06 | 84.88 |
| JointEnergy | 56.52 | 87.02 | 88.14 | 61.80 | 72.26 | **82.90** | 53.41 | 92.65 | 87.85 | 56.42 | 80.63 | 85.44 |
| GNNSAFE | 45.24 | 88.19 | 87.90 | **55.01** | **73.19** | 82.54 | 30.19 | 95.26 | 91.06 | 46.53 | 82.34 | 85.74 |
| Ours | **43.41** | **89.05** | **89.34** | 55.73 | 73.04 | 81.77 | **28.64** | **95.68** | **91.60** | **45.78** | **82.47** | **85.82** |

in the average FPR and a 2.32% increase in the average AUROC compared to GNNSAFE. To assess the robustness of our method, we vary the ID and OOD labels of DBLP, designating one label as OOD class and other labels as ID class, and the results are summarized in Table 4. It can be observed that our method achieves the best results in most cases, underscoring the stability of our method.

**Evaluation on Method Components.** We assess the efficacy of the components of our proposed

Table 5: Evaluation of Method Components on OOD detection.

| Method | Facebook FPR ↓ | Facebook AUROC ↑ | Yeast FPR ↓ | Yeast AUROC ↑ | Movie FPR ↓ | Movie AUROC ↑ | DBLP FPR ↓ | DBLP AUROC ↑ | BlogCatalog FPR ↓ | BlogCatalog AUROC ↑ | Average FPR ↓ | Average AUROC ↑ |
|---|---|---|---|---|---|---|---|---|---|---|---|---|
| Ours(w/o LPM) | 8.81 | **95.75** | 81.61 | **79.27** | 65.88 | 67.80 | 52.59 | 86.23 | 89.87 | 65.49 | 59.75 | 78.91 |
| Ours(w/o FPM) | 7.86 | 94.77 | 88.41 | 77.12 | 60.59 | 69.38 | **41.93** | **89.59** | 88.62 | 68.47 | 57.48 | 79.87 |
| Ours(w/o Propagation) | 9.12 | 95.34 | 82.53 | 78.86 | 65.88 | 66.68 | 53.06 | 88.34 | 89.97 | 67.18 | 60.11 | 79.28 |
| Ours | **7.55** | 95.15 | **74.50** | 78.79 | **56.76** | **71.17** | 43.41 | 89.34 | **86.40** | **69.83** | **53.72** | **80.86** |

method for OOD detection performance. Comparisons are conducted on multiple datasets, between the version of our method with and without feature pattern matching (FPM), or label pattern matching (LPM) as well as propagation, and the results are summarized in Table 5. The results demonstrate that, surprisingly, for certain metrics on some datasets, the versions without LPM, FPM, or propagation performed better. However, overall, these components positively contributed to OOD detection performance on most datasets. For instance, although the AUROC metric on the Facebook dataset improved by 0.6% without label pattern matching, leveraging label pattern matching reduced the average FPR by 6.03% across five datasets. Additionally, feature pattern matching and propagation respectively decreased the average FPR by 3.76% and 6.39% across the same datasets. These highlight the effectiveness of leveraging pattern information from the training data and structural information of graph for enhancing OOD detection on multi-label node classification in graph.

## 5.3 ABLATION STUDY

We proceed to conduct a detailed analysis of the effect of hyper-parameters and encoder backbones to gain deeper insights into the practice and application of our proposed method. All ablations are performed on the DBLP dataset.

**Effect of Feature Pattern Matching $P$ and $O$.** In Figure 4, we evaluated the impact of different $P$ and $O$ combinations on OOD detection performance in feature pattern matching. The results indicate that for various $O$ values, both too small or too large $P$ values lead to performance deterioration,

with an optimal range of 1000-1500. For the dblp dataset, when $P$ is between 1000-1500, the larger $O$ value significantly outperforms smaller ones. Notably, due to the varying number of nodes across datasets, the optimal $O$ value differs. In our experiments, we fixed $P$ at 1000 for all datasets. The $O$ values were set to 7000 for dblp, 100 for Facebook and Yeast, and 1000 for Movie and BlogCatalog.

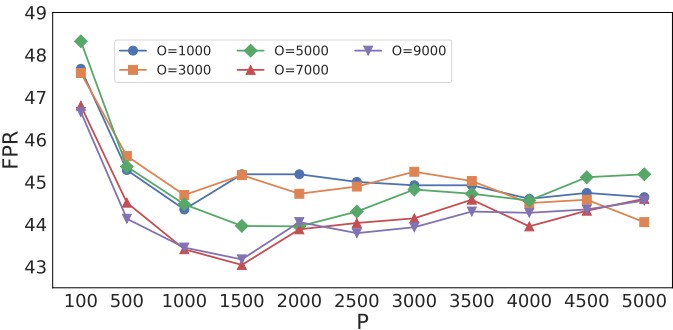

Figure 4: Ablation study on feature pattern matching parameters $P$ and $O$.

**Effect of Propogation Weight $\alpha$ and Iterations $T$.** In Figure 5, we systematically examine the effect of the propagation weight $\alpha$ and the iteration steps $T$. We vary the size of $\alpha$ uniformly from 0.1 to 0.9 and set $T = \{1, 2, 4, 8, 16, 32, 64\}$. Several observations emerge: (1) With small $\alpha$ values (e.g. 0.1–0.5), the FPR tends to increase as $T$ increases. However, with larger $\alpha$ values (e.g. 0.7–0.9), the FPR initially decreases and then increases with increasing $T$. Overall, for all $\alpha$ values, when $T$ exceeds 8, the FPR exhibits a significant increase with further increments in $T$. (2) When the T value is higher (e.g. greater than 4), the FPR decreases as the $\alpha$ value increases, and when the $T$ value is small, the middle sized $\alpha$ value corresponds to a lower FPR as the $T$ value decreases. These observations lead us to opt for a balanced configuration, specifically $\alpha = 0.5$ and $T = 1$, in our implementation achieving superior performance in OOD detection.

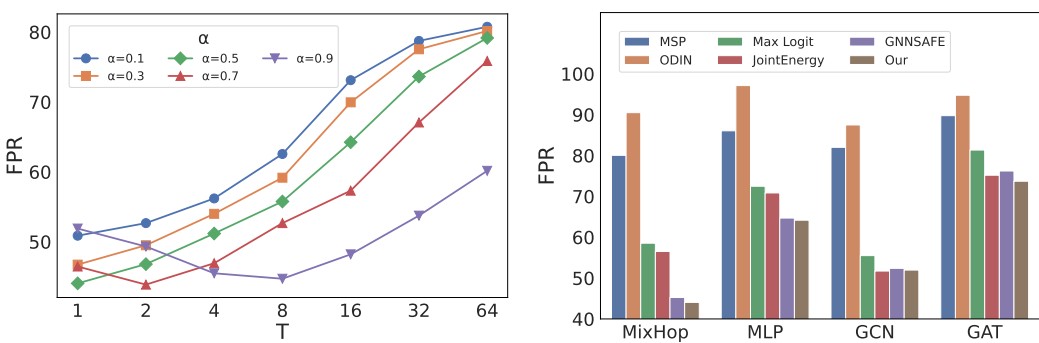

Figure 5: Ablation study on propagation parameters $\alpha$ and $T$.

Figure 6: Ablation study on network backbones with different.

**Effect of Network Backbones.** In Figure 6, we present the results of our method and its competitors using various network backbones, including MixHop, MLP, GCN and GAT, follow in the setup in GNNSAFE (Wu et al., 2023). The results demonstrate the robust performance of our method across different backbones, consistently surpassing all other baseline methods in terms of FPR.

## 6 CONCLUSION

In this paper, we formulate the problem of OOD detection for multi-label graph node classification, and propose a pattern matching-based OOD detection method. Our method first utilizes the logit and feature information of the data to match with the patterns in the training data, resulting in a matching score. Then, we employ a propagation algorithm to further amplify the difference in matching scores between ID and OOD samples. By normalization and summing two scores, we obtain the final confidence score, which is used to determine whether the sample belongs to the ID or OOD category. Extensive experiment results demonstrate the superiority of our proposal.

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

# A APPENDIX

## A.1 DATASETS

**Datasets and Splits**

The datasets used to validate our proposal are required to be graph-structured and multi-labeled. Following the setting in literature Zhou et al. (2021), our experiments are based on five well-established real-world datasets that serve as standard benchmarks for node classification: Facebook, Yeast, Movie, DBLP and BlogCatalog. **Facebook** Leskovec & Mcauley (2012). This dataset consists of several ego networks, where circles of different nodes serve as labels, allowing nodes to be assigned to multiple circles. Node attributes, such as education status and working positions, are derived from personal social tags. In our experiments, we select one typical ego network with 792 nodes. For OOD detection setting, the first 9 labels are considered as ID labels, while the last 8 labels are designated as OOD labels.

**Yeast** Cheng et al. (2002). This dataset contains diverse details about genes of a specific organism, where each gene is treated as a node, and their interactions form a graph. The labels correspond to the functions performed by the proteins encoded by the genes. In our experiments, 4 labels are designated as ID labels, while the remaining labels serve as OOD labels.

**Movie** Leskovec & Mcauley (2012). The dataset encompasses individual ratings and tags related to movies. A movie is considered a node in a graph, and a link is established between two movies if they share a common director. The original dataset's movie tags are utilized as attributes, while the movie genres are regarded as the labels. We choose 13 out of 20 labels as ID.

**DBLP** Akujuobi et al. (2019). As a multi-label citation dataset, features nodes represent authors and edges denote co-authorship relations. The dataset comprises four labels, each signifying a distinct research area: database (DB), data mining (DM), artificial intelligence (AI), and information retrieval (IR). DM, AI and IR are designated as ID labels, while DB serves as the OOD label. Author attributes consist of the concatenated text from the titles of papers published by the author.

**BlogCatalog** Chen et al. (2018) Each node represents a blogger on the BlogCatalog website, and edges connect bloggers who are friends. The labels indicate the categories where a blogger publishes. The dataset contains no node attributes. We utilize node embeddings generated by Node2Vec[2] in our experiments as node attributes, resulting in 39 labels. For OOD detection setting, the first 25 labels are designated as ID labels, while the last 14 labels serve as OOD labels.

## A.2 PSEUDO CODE

---

[2]https://www.csie.ntu.edu.tw/ cjlin/libsvmtools/datasets/multilabel.html

---

**Algorithm 1** Pattern Matching Algorithm for OOD-MLNC

---

**Input**: Graph $G$ with label matrix $\mathbf{Y}$, node feature matrix $\mathbf{X}$, and adjacency matrix $\mathbf{A}$.
**Parameter**:Intervals number $P$, Samples number threshold $O$, Logit threshold $\tau$, iteration steps $T$, weight $\alpha$, network training parameters $\theta$.
**Output**: Confidence score $\tilde{\mathbf{S}}$.

1: Initialize $\mathbf{Z}^{(0)} = \mathbf{X}$
2: **while** it not converged **do**
3: $\quad \mathbf{Z}^{(L)}, \mathbf{Z}^{(L-1)} \leftarrow \text{GNN}(\mathbf{X}, \mathbf{Y}_{tr}, \mathbf{A}|\theta)$
4: **end while**
5: Extract a sub-matrix $\mathbf{Y}_{tr}$ and $\mathbf{Z_{tr}^{(L-1)}}$ only with labeled nodes from label matrix $\mathbf{Y}$ and feature matrix $Z^{(L-1)}$
6: $\mathbf{M}^+, \mathbf{M}^- \leftarrow count(\mathbf{Y}_{tr})$
7: **while** for each $z_i$ in $\mathbf{Z}^{(L-1)}$ **do**
8: $\quad \Phi_{in}^k(z_{ik}; O) = \frac{1}{N_{tr}} \min(count(I_j|z_{ik} \in I_j), O)$
9: $\quad \mathbf{s_f} = \frac{1}{K_f} \sum_{k=1}^{K_f} \Phi_{in}^k(z_{ik}; O)$
10: **end while**
11: **while** for each $z_{ij}$ in $\mathbf{Z}^{(L)}$ **do**
12: $\quad$ *if* $z_{ij} < \tau$ *then* $e_{ij} \leftarrow 0$; *else* $e_{ij} \leftarrow z_{ij}$
13: **end while**
14: **while** for each node truncate logit $\mathbf{e_i}$ in $\mathbf{E}$ **do**
15: $\quad$ *if* $\mathbf{e}_{ij} > 0$ *then* $\mathbf{M}_{ij} \leftarrow \mathbf{M}_{ij}^+$; *else* $\mathbf{M}_{ij} \leftarrow \mathbf{M}_{ij}^-$
16: $\quad \mathbf{s_l} \leftarrow \mathbf{e_i} \cdot \mathbf{M}$
17: **end while**
18: **while** iterations not up to T **do**
19: $\quad \mathbf{S} \leftarrow \alpha\mathbf{S} + (1-\alpha)\mathbf{D}^{-1}\mathbf{A}\mathbf{S}$
20: **end while**
21: Normalization of $\mathbf{S_f}$ and $\mathbf{S_l}$
22: $\tilde{\mathbf{S}} = \mathbf{S_f} + \mathbf{S_l}$
23: **return** $\tilde{\mathbf{S}}$

---

