# OpenReview forum: "PATTERN MATCHING-BASED OUT-OF-DISTRIBUTION DETECTION FOR MULTI-LABEL NODE CLASSIFICATION"
_ICLR.cc/2025/Conference — Submitted to ICLR 2025_

### Official Review · Reviewer_5Z1x · 2024-10-16

**Soundness:** 3
**Presentation:** 2
**Contribution:** 3
**Rating:** 3
**Confidence:** 2

**Summary:**

This manuscript studies the OOD problem on multi-label graphs. The authors introduce feature pattern matching, label pattern matching, and label propagation via GNN and then conduct the multi-label OOD classification on nodes. Experimental results, as they reported, demonstrate the effectiveness of their solution.

**Strengths:**

S1: the technical solution seems to make sense.

S2: Figure 1 is beautiful

**Weaknesses:**

W1: code not available

W2: I want to know the difference in MLNC OOD detection between graph and CV/NLP data. My main concern is that there seems no difficulty in applying the existing techniques on the graph. I wonder about the unique challenges of graph domains and how the authors solve them with their creative contributions.

W3: It seems that this paper is not the first work to solve multi-label graph OOD detection. See this work: Evidence-Based Out-of-Distribution Detection on Multi-Label Graphs.

W4: I think the paper could be further polished. The language writing could be significantly improved.

W5: It would be more solid if the proposed solutions could have some theoretical guarantee.


Overall, this manuscript may propose a self-contained solution to multi-label OOD problems on graph domains. Although I hate to say this, I am afraid I have to say (with full respect) that the novelty and contributions of this submission might be not strong enough for this conference. I think it might be further enhanced with some theoretical support, more convincing public code, and more clear paper writing

Plus: Kindly note that when I say "more clear writing", I do not only hope the language issues could be improved (e.g. line 075-077) but I also think there are some logical points in the paper should be polished. For example, the authors use a large part of the content to say they use sigmoid instead of softmax for multi-label problems. Do the authors wish to treat this as their contribution? because I think it is such a simple and obvious thing. If the authors wish to claim this tiny change matters and is a significant technique contribution, maybe they can provide with in-depth theory analysis further. Many other places are looked like just for filling the page length but I don't know why they are necessary to be mentioned. The authors present some technique choices and different results (like Figure 3) but we need to know why.

**Questions:**

Q1: between ID and OOD nodes, there seem to be some middle cases, for example, ID nodes have labels $\{A, B, C\}$ with distribution $P_1(A, B, C)$, some other nodes have labels $\{A, B, C, D, E\}$ with a totally different distribution $P_2(A,B,C,D,E)$, and then the rest node have labels $\{D, E\}$ with the distribution $P_3(D,E)$. I wonder how to deal with $P_1$ and $P_2$.

Q2: refer to W2

Q3: refer to W5

---

> ### Author Response · Authors · 2024-11-23
> **Thank you; more discussion**
>
> **Q1:** code not available.
>
> **A1:** We will open-source the code after the publication of the paper.
>
> **Q2:** I want to know the difference in MLNC OOD detection between graph and CV/NLP data. My main concern is that there seems no difficulty in applying the existing techniques on the graph. I wonder about the unique challenges of graph domains and how the authors solve them with their creative contributions.
>
> **A2:** In the MLNC OOD detection problem, the primary challenge lies in the fact that each node can be assigned multiple labels. As shown in Figure 3, methods like MSP, which rely solely on the maximum confidence score, can be adapted but fail to achieve satisfactory performance. To address this, our label pattern matching method incorporates information from multiple labels. Furthermore, we leverage the crucial label dependency relationships inherent in multi-label scenarios, a factor not considered in other OOD detection methods. Inspired by the VIM method, we also integrate the feature distribution information from the training set to enhance the robustness of our method. Finally, while CV/NLP datasets typically assume independent and identically distributed data, graph data contains exploitable topological structures. Therefore, similar to GNNASAFE, we employ a label propagation algorithm to further refine the scores.
>
> **Q3:** It seems that this paper is not the first work to solve multi-label graph OOD detection. See this work: Evidence-Based Out-of-Distribution Detection on Multi-Label Graphs.
>
> **A3:** Thank you for pointing this out. Based on our understanding, the problem definition in that work differs significantly from ours. Specifically, in [1], a node is defined as an OOD sample if it contains any unknown category. In contrast, under our definition, a node is classified as OOD only if it does not belong to any known category. Our definition is a natural extension of the one proposed in JointEnergy.
>
> [1] Evidence-Based Out-of-Distribution Detection on Multi-Label Graphs
>
> **Q4:** I think the paper could be further polished. The language writing could be significantly improved.
>
> **A4:** Thank you for your suggestion regarding our writing. Our intention was to illustrate that, compared to directly using logit information, utilizing probabilities mapped through the sigmoid function yields less effective results for the OOD detection problem. We sincerely apologize for the confusion caused by the unclear logic in this section. Based on your suggestion, we have revised and condensed this part in the original manuscript.
>
> **Q5:** It would be more solid if the proposed solutions could have some theoretical guarantee.
>
> **A5:** Similar to works like MSP and Max Logit, our method is heuristic-based and validated through experimental results. Specifically, our method is designed for the graph multi-label OOD scenario, leveraging the unique characteristics of this problem to develop a tailored solution. Therefore, as demonstrated by the experimental results, I believe our method is better suited for the MLNC OOD problem compared to other approaches.
>
> **Q6:**  between ID and OOD nodes, there seem to be some middle cases, for example, ID nodes have labels $A, B, C$ with distribution $P_1(A, B, C)$, some other nodes have labels $A, B, C, D, E$  with a totally different distribution $P_2(A, B, C, D, E)$, and then the rest node have labels $D, E$ with the distribution $P_3(D, E)$. I wonder how to deal with
> $P_1$ and $P_2$.
>
> **A6:** In fact, such nodes do exist. We define both $P_1$  and $P_2$  as ID nodes since they have known labels $A$, $B$ and $C$, while $P_3$ is a typical OOD node because it does not belong to any known category. Since we cannot determine during the training phase which nodes belong to $P_2$ , no additional processing is performed. However, if there are a large number of $P_2$ nodes in the training set, it can significantly increase the difficulty of detecting $P_3$ nodes during the OOD detection process (as discussed in response to Reviewer 1's Q3 regarding why DBLP performs poorly in certain domains). We will provide a more comprehensive discussion of this issue in future revisions.

---

> > ### Comment · Reviewer_5Z1x · 2024-11-27
> >
> > Thank the authors for their reply. This message is to confirm that I have read your response and I am engaged in the discussion, as required by ICLR. I carefully read the message and I will keep my score unchanged. I think the authors might also realize the reasonability of my score from their response.
> >
> >
> > KInd regards,

---

### Official Review · Reviewer_2CTU · 2024-11-04

**Soundness:** 3
**Presentation:** 3
**Contribution:** 3
**Rating:** 6
**Confidence:** 3

**Summary:**

This paper addresses the challenge of out-of-distribution (OOD) detection for graph neural networks (GNNs) in multi-label classification tasks. The authors propose a novel, pattern matching-based method that combines feature and label pattern matching to produce two matching scores. These scores are further refined with topological structure adjustments to derive confidence scores, indicating the likelihood of a test sample being an OOD instance.

**Strengths:**

1. The multi-label classification on graph OOD detection is important and interesting.
2. The research problem is challenging.
3. The method is supported by theoretical analysis, which strengthens the validity of the approach.
4. The proposed method demonstrates promising results across multiple datasets, including a real-world application, highlighting its practical effectiveness and versatility.

**Weaknesses:**

1. The connection between pattern matching and multi-label classification needs clearer illustration. Specifically, it would be helpful to explain how pattern matching contributes to or enhances the multi-label classification process and to outline the role it plays within the proposed framework.

2. The scalability of the proposed method is not thoroughly explored, particularly for large-scale graphs where such an approach might be impractical.

**Questions:**

same as in the Weakness

---

> ### Author Response · Authors · 2024-11-23
> **Thank you; more discussion**
>
> **Q1:** The connection between pattern matching and multi-label classification needs clearer illustration. Specifically, it would be helpful to explain how pattern matching contributes to or enhances the multi-label classification process and to outline the role it plays within the proposed framework.
>
> **A1:** We argue that the goal of OOD detection is not to enhance classifier accuracy but to increase their trustworthiness. Our method is a post-processing method that computes feature pattern matching scores and label pattern matching scores based on the model's output features and logits, respectively, to quantify the differences between test samples and training samples (i.e., ID samples). Compared to other methods, our method achieves better performance because it is more tailored to the graph-based multi-label OOD detection scenario. Specifically, it leverages label dependencies and graph topology information, and the incorporation of multi-source information further enhances its robustness.
>
> **Q2:** The scalability of the proposed method is not thoroughly explored, particularly for large-scale graphs where such an method might be impractical.
>
> **A2:** As discussed in Q5 with Reviewer 2, the time complexity of our method is $O(({K_l}^2+K_f) \cdot N)$, which ensures good scalability and allows it to handle large-scale graphs effectively.

---

### Official Review · Reviewer_bjgR · 2024-11-04

**Soundness:** 2
**Presentation:** 2
**Contribution:** 3
**Rating:** 5
**Confidence:** 3

**Summary:**

Multi-label classification on graphs is an important task. This paper studies the multi-label graph OOD detection problem and proposes a simple yet effective pattern matching-based OOD detection method. Experimental results show that the method proposed in this paper is good.

**Strengths:**

1. This paper is easy to follow.
2. Multi-label classification on graphs is an important issue.
3. Improvement in experiments seems good.

**Weaknesses:**

1. The method proposed in this paper lacks necessary theoretical analysis.

2. The idea of confidence score is not novel, as it has been widely used in works related to OOD and class imbalance.

3. I noticed that the authors cited the work of Zhao et al.[1], but they did not use the relevant new datasets in the experimental section.

4. The ablation experiments show it is very sensitive to hyperparameters, especially regarding label propagation.

5. No time complexity or runtime comparison. No pseudo-code and no data statistics.

   [1]Multi-label node classification on graph-structured data. TMLR 2023.

**Questions:**

Can you provide the experimental results related to the PCG, HumLoc, and EukLoc datasets [1] ?

---

> ### Author Response · Authors · 2024-11-23
> **Thank you; more discussion [1/2]**
>
> **Q1:** The method proposed in this paper lacks necessary theoretical analysis.
>
> **A1:** Similar to works like MSP and Max Logit, our method is heuristic-based and validated through experimental results. Our method is specifically designed for the multi-label graph OOD detection scenario, leveraging multi-source information to address this challenge. More specifically, our method incorporates feature distribution information, label dependencies, and graph topology information based on the characteristics of the problem. As demonstrated by the experimental results, we believe our method is better suited for this task.
>
> **Q2:** The idea of confidence score is not novel, as it has been widely used in works related to OOD and class imbalance.
>
> **A2:** We use the term confidence score to align with its usage in other works, making it easier for peers to understand and reproduce our work. However, the technical details for obtaining the confidence score in our method differ from those in other studies. For example, the MSP method computes confidence scores by applying softmax to the model’s logit outputs and selecting the maximum value. In contrast, our method leverages multi-source information and accounts for label dependencies and graph topology relationships in the context of multi-label graph OOD detection. Therefore, although both our method and others ultimately rely on confidence scores to detect OOD samples, our method is better suited for multi-label graph OOD detection, as supported by the experimental results.
>
> **Q3:** I noticed that the authors cited the work of Zhao et al.[1], but they did not use the relevant new datasets in the experimental section.
>
> **A3:** According to your request, we have supplemented the experimental results related to the PCG, HumLoc, and EukLoc datasets. Experimental results indicate that, overall, our method still demonstrates a performance advantage.
>
> | Method        | PCG             |               | HumLoc         |               | EukLoc         |               | Average        |               |
> |---------------|-----------------|---------------|----------------|---------------|----------------|---------------|----------------|---------------|
> |               | FPR↓            | AUROC↑        | FPR↓           | AUROC↑        | FPR↓           | AUROC↑        | FPR↓           | AUROC↑        |
> | MSP           | 94.48           | 48.89         | 96.56          | 57.64         | 91.02          | 54.21         | 94.02          | 53.58         |
> | ODIN          | 94.48           | 48.90         | 96.88          | 57.65         | 91.02          | 54.20         | 94.12          | 53.58         |
> | Max Logit     | 93.00           | 52.26         | 96.25          | 60.15         | 91.30          | 56.23         | 93.51          | 56.21         |
> | JointEnergy   | 93.55           | 53.71         | 95.00          | 60.73         | 92.53          | 56.58         | 93.69          | 57.00         |
> | GNNSAFE       | 93.00           | **54.45**         | 85.62          | 70.27         | 91.49          | 62.19         | 90.03          | 62.30         |
> | Ours          | **92.82**           | 51.73         | **80.31**          | **73.01**         | **87.81**          | **67.35**         | **86.98**          | **64.03**         |
>
> **Q4:** The ablation experiments show it is very sensitive to hyperparameters, especially regarding label propagation.
>
> **A4:** From the results of the ablation experiments, our method does show sensitivity to the propagation parameter T. For multiple datasets, the ablation experiments reveal a consistent pattern: excessively large T values lead to a sharp decline in OOD detection performance. The underlying reason is that excessive propagation results in over-smoothing, where the information from each node spreads across the entire graph, causing the loss of node-specific information and making nodes harder to distinguish. As a result, we limit the propagation to a single round when selecting the propagation hyperparameter. Thank you for pointing this out, we will provide a more detailed explanation of this phenomenon in the revised version.

---

> > ### Comment · Reviewer_bjgR · 2024-11-27
> > **Respone to the rebuttal**
> >
> > Thank the authors for their reply. My concerns are partially addressed. I have gone through all the comments and suggest the authors incorporating the  feedbacks of the reviews and rebmit  the following conference.

---

> ### Author Response · Authors · 2024-11-23
> **Thank you; more discussion [2/2]**
>
> **Q5:** No time complexity or runtime comparison. No pseudo-code and no data statistics.
>
> **A5:** In our method, the time complexity primarily comes from the computation of feature pattern matching and label pattern matching. During the computation of feature pattern matching, each sample needs to query the pre-modeled density function for each of its $K_f$-dimensional features. Thus, the time complexity for feature pattern matching is $O(K_f)$ per sample. For label pattern matching, each sample's logit vector with $1 \times K_l$ dimension needs to perform a dot product with an $M_{K_l \times K_l}$ matrix. The result is in a time complexity of $O({K_l}^2)$ per sample. Therefore, for $N$ test samples, the overall time complexity of our method is:  $O(({K_l}^2+K_f) \cdot N)$.
>
> Due to the limits of page, we included the pseudocode in the appendix when submitting the paper, and the data statistics you mentioned are presented in Table 2 of the main text.

---

### Official Review · Reviewer_uCQL · 2024-11-05

**Soundness:** 3
**Presentation:** 3
**Contribution:** 2
**Rating:** 5
**Confidence:** 4

**Summary:**

This paper focuses on detecting OOD in graph multi-label classification problems. It defines the problem, and proposes a method to quantify feature pattern matching and label pattern matching, utilizing label propagation to derive the final confidence score for OOD. Empirical evaluations on five real-world datasets demonstrate that the proposed method achieves better average accuracy compared to selected baselines.

**Strengths:**

Strengths:
1. The paper studies OOD in a practical setting, which could lead to potential real-world application and inspire further research and development in the field.

2. The proposed method is straightforward and easily understandable.

3. Ablation studies are included to analyze the contribution of different components of the proposed method.

**Weaknesses:**

Weaknesses:
1. The paper lacks details on the normalization of two scores in Equation 11.  Additionally, the equal weighting of these scores seems arbitrary and might need further investigation into alternative weighting schemes or a data-driven approach to determine optimal weights.

2. Figure 3 in the paper lacks explicit details about its origin. It is unclear whether the figure is based on specific data analysis or merely serves as a random illustration. This lack of clarity is particularly crucial because the figure is used to explain the design choice of "sum logits" in the absence of theoretical analysis.

3. Though the experimental results show the proposed method achieves a better score on average, it is suggested to provide further analysis to explain why it performs worse in certain domains within the DBLP dataset. In addition, the paper lacks a theoretical foundation to support the effectiveness of the proposed method.

4. The major concern is the limited novelty of the proposed method. Feature pattern matching and label pattern matching have been widely adopted in previous works like NAC and (Hendrycks et al., 2022). And label propagation is also common in handling graph data to refine pseudo labels or predicted scores. The proposed method appears to be a direct combination of existing methods in multi-label classification OOD and graph mining, with limited novel contributions.

**Questions:**

See Weaknesses

---

> ### Author Response · Authors · 2024-11-23
> **Thank you; more discussion [1/2]**
>
> **Q1:** The paper lacks details on the normalization of two scores in Equation 11. Additionally, the equal weighting of these scores seems arbitrary and might need further investigation into alternative weighting schemes or a data-driven method to determine optimal weights.
>
> **A1:** Regarding the details of the normalization function: given the significant difference in scales between the two score values and the need to preserve the distribution characteristics of the scores, we chose the commonly used Z-Score normalization for the regularization process.
>
> Additionally, we have included ablation experiments on the weights. Spcifically, we assign weights $\beta$ and $1-\beta$ to $S_f$ and $S_l$, respectively. The experimental results indicate that no single weighting achieves optimal performance across all datasets. We believe the reason lies in the fact that different datasets require different types of information, leading to variations in suitable weights. For instance, assigning a higher weight to the label pattern matching score yields better results on the DBLP dataset. However, when the weights are skewed toward extreme values, the average performance of OOD detection declines. This suggests that over-reliance on a single source of information negatively impacts multi-label graph OOD detection, which is precisely why we propose incorporating multi-source information.
>
> In the absence of sufficient prior knowledge to determine which type of information is better suited for a given dataset, we default to assigning equal weights to both scores. Your suggestion is highly valuable, and we will include a discussion on this aspect in the revised version.
>
> | β   | Facebook       |               | Yeast         |               | Movie         |               | DBLP          |               | BlogCatalog    |               | Average       |               |
> |-----|----------------|---------------|---------------|---------------|---------------|---------------|---------------|---------------|----------------|---------------|---------------|---------------|
> |     | FPR↓           | AUROC↑        | FPR↓          | AUROC↑        | FPR↓          | AUROC↑        | FPR↓          | AUROC↑        | FPR↓           | AUROC↑        | FPR↓          | AUROC↑        |
> | 0.0 | 8.49           | 94.39         | 81.30         | 76.50         | 60.59         | 69.97         | 41.49         | 90.08         | 89.68          | 67.61         | 56.31         | 79.71         |
> | 0.1 | 8.49           | 94.54         | 81.30         | 77.05         | 59.12         | 71.20         | 41.75         | 90.12         | 86.31          | 69.13         | 55.39         | 80.40         |
> | 0.3 | 7.55           | 94.66         | 79.91         | 77.91         | 55.29         | 72.29         | 41.59         | 90.03         | 85.82          | 69.97         | 54.03         | 80.97         |
> | 0.5 | 7.55           | 94.81         | 71.10         | 78.46         | 56.76         | 72.50         | 43.11         | 89.65         | 86.79          | 69.76         | 53.06         | 81.03         |
> | 0.7 | 7.23           | 94.92         | 68.78         | 78.89         | 58.24         | 72.52         | 44.39         | 88.86         | 86.79          | 69.25         | 53.08         | 80.88         |
> | 0.9 | 7.23           | 95.07         | 70.94         | 79.14         | 57.94         | 72.51         | 49.16         | 87.43         | 87.27          | 68.65         | 54.50         | 80.56         |
> | 1.0 | 7.23           | 95.34         | 78.83         | 79.06         | 57.65         | 72.50         | 53.32         | 86.38         | 87.27          | 68.31         | 56.86         | 80.31         |
>
> **Q2:** Figure 3 in the paper lacks explicit details about its origin. It is unclear whether the figure is based on specific data analysis or merely serves as a random illustration. This lack of clarity is particularly crucial because the figure is used to explain the design choice of "sum logits" in the absence of theoretical analysis.
>
> **A2:** Regarding the choice of sum logit, we conducted validation experiments on multiple graph datasets and consistently observed the same conclusion: sum logit outperforms other methodes. However, due to space constraints, we only present the results from the DBLP dataset in the paper to illustrate this phenomenon.

---

> ### Author Response · Authors · 2024-11-23
> **Thank you; more discussion [2/2]**
>
> **Q3:** Though the experimental results show the proposed method achieves a better score on average, it is suggested to provide further analysis to explain why it performs worse in certain domains within the DBLP dataset. In addition, the paper lacks a theoretical foundation to support the effectiveness of the proposed method.
>
> **A3:** Similar to works like MSP and Max Logit, our method is heuristic-based and validated through experimental results, and here is the analysis of why our method performs poorly in certain domains of the DBLP dataset: In the DBLP dataset, among all nodes with the DM label, 39\% also have additional labels, which is the highest proportion compared to other categories (DB, AI, and IR have proportions of 26\%, 17\%, and 36\%, respectively). Consequently, when the DM category is treated as the OOD category, the training set still contains many nodes that actually have the DM label (i.e., nodes labeled with DM along with other labels), allowing the model to learn from these nodes. During the testing phase, the model predicts OOD nodes to have similar label patterns and feature patterns to these nodes, leading to a decline in the performance of our method.
>
> **Q4:** The major concern is the limited novelty of the proposed method. Feature pattern matching and label pattern matching have been widely adopted in previous works like NAC and (Hendrycks et al., 2022). And label propagation is also common in handling graph data to refine pseudo labels or predicted scores. The proposed method appears to be a direct combination of existing methods in multi-label classification OOD and graph mining, with limited novel contributions.
>
> **A4:** We argue that there exist significant difference between our work and existing works utilizing feature and logit information for addressing the OOD problem. First, our problem definition is different. These methods are designed for single-label OOD detection or multi-label OOD detection in non-image data, whereas we are focused on solving the graph multi-label OOD problem.
>
> Second, our technical details also differ significantly. For example, NAC processes the network’s multi-layer outputs from the perspective of activation values, whereas we simply use the output from the penultimate layer to represent the feature distribution. Moreover, leveraging the multi-label characteristics of the problem, we propose using label correlations to calculate the label pattern matching score, which has not been addressed in existing OOD detection methods. Experimental results further demonstrate that our method achieves superior performance in graph multi-label OOD scenarios.

---

### Meta-Review · Area_Chair_AcPZ · 2024-12-17

**Metareview:**

This paper studies the graph OOD problem which is encountered when employing graph neural networks (GNNs) in real-world scenarios.  The authors propose an approach which adapts the graph OOD problem which is typically on a single-label classification setting to a multi-label one.  Moreover, the authors propose an approach to combine feature pattern matching and label pattern matching to estimate confidence of samples being OOD.

Reviewers pointed out several weaknesses in this work which led to a reject recmomendation at this time.  There were a few themes of concerns raised:

- Perceived limited novelty, given the previous approaches of feature pattern matching and label pattern matching have been adopted in prior works (ucQL, 5Z1x)

- Lack of theoretical foundation to support effectiveness of the method; the method is largely heuristic (uCQL, 5Z1x, bjgR)

- Missing experiments around runtime and scalability (2CTU, bjgR)

I encourage the authors to fold these into their revision.

**Additional Comments On Reviewer Discussion:**

Authors engaged with the reviewers concerns during rebuttal and discussion phase.  In response to uCQL around score normalization, authors showed experiments demonstrating that different weighting strategies are optimal for each dataset and no unified observation can be drawn here.  Moreover, in response to bjgR, authors included experiments on multiple new datasets including PCG, HumLoc and EukLoc which demosntrate a retained performance advantage.  The authors also included pseudocode and some discussion around time complexity.  Reviewers remained mostly unconvinced that the work was ready for this conference, and encouraged the authors to resubmit after revising.

---

### Decision · Program_Chairs · 2025-01-22

Reject